

# Acute net stress of young adult zebrafish (*Danio rerio*) is not sufficient to increase anxiety-like behavior and whole-body cortisol

Amy Aponte and Maureen L. Petrunich-Rutherford

Department of Psychology, Indiana University Northwest, Gary, IN, United States of America

## ABSTRACT

In recent years, the zebrafish (*Danio rerio*) has become a popular model to study the mechanisms of physiological and behavioral effects of stress, due to the similarity in neural structures and biochemical pathways between zebrafish and mammals. Previous research in this vertebrate animal model has demonstrated an increase in whole-body cortisol resulting from an acute (30-second) net handling stress, but it remains unclear whether such a stressor will concomitantly increase anxiety-like behavior. In addition, as the previous study examined the effects of this acute stressor in adult zebrafish after a brief period of isolation, it is unclear whether this stressor would be effective in eliciting cortisol increases in younger aged subjects without isolation. In the current study, young adult zebrafish (approximately 90 days post-fertilization) were briefly exposed to a net handling stressor and were subsequently subjected to either the novel tank test or the light/dark preference test. The novel tank test was used to measure exploration and habituation in response to a novel environment, and the light/dark preference test was used to measure locomotor activity and scototaxis behavior. All subjects were sacrificed 15 minutes post-stressor and were analyzed for whole-body levels of cortisol. Contrary to expectations, there was no effect of acute net handling on cortisol levels. Similarly, acute net handling did not significantly induce anxiety-like behavior during the novel tank test or the light/dark preference test. Our findings demonstrate that there are possible developmental differences in response to an acute net handling stress, as we did not observe alterations in hormonal or behavioral measures of anxiety in young adult zebrafish. Alternatively, if zebrafish are not isolated before the stressor, they may be more resilient to a brief acute stressor. These results suggest the need for a different or more intense acute stressor in order further explore neuroendocrine mechanisms and anxiety-like behavior at this developmental stage in the zebrafish animal model.

Corresponding author
Maureen L. Petrunich-Rutherford,
mlpetrun@iun.edu

# INTRODUCTION

Among psychiatric disorders, anxiety disorders are the most common in adults in the United States, with a lifetime prevalence rate at an estimated 28.8% (*Kessler et al., 2005*). In addition to immediate and potentially harmful long-term health effects, anxiety disorders present an

annual medical cost of 42.3 to 46.6 billion dollars in the United States (*DuPont et al., 1996*). In an attempt to address both individual health concerns and ease the economic burden, research in anxiety has focused on investigating the neural and endocrine mechanisms associated with anxiety to better understand the pathology and treatment options associated with stress-related conditions.

Brain structures such as the hypothalamus, amygdala, and hippocampus are responsible for controlling and mediating the effects of stress (*McEwen, 2007*). When confronted with a threatening situation, an organism experiences an innate stress response, consisting of a series of physiological and behavioral changes that serve as coping mechanisms to ultimately return the organism to a homeostatic state (*Gold, 2015*). The autonomic nervous system and hypothalamic-pituitary-adrenal (HPA) axis serve important roles in regulating stress responses. Alterations in the structure and/or function of the hypothalamus and the HPA axis are associated with anxiety and other stress-related conditions (*Faravelli et al., 2012*; *Terlevic et al., 2013*; *Zorn et al., 2017*). Studying the effects of stress on physiology and behavior is, therefore, imperative to understanding and treating anxiety disorders.

Animal models are critical for studying the behavior, neural, and endocrine responses to stress exposure. Among vertebrates, the zebrafish (*Danio rerio*) is increasingly becoming a model organism in biomedical research. In particular, the zebrafish model is rapidly gaining popularity in neuroscience, largely because it requires inexpensive, simple handling, and breeding is rapid and richly produces offspring (*Gerlai, 2010*). Most importantly, examination of central nervous system anatomy, genomic sequences, and biochemical pathways in zebrafish have been found to have similar mammalian homologs and further supports the use of zebrafish as a model organism for studying neural mechanism of behaviors (*Guo, 2009*; *Holzschuh et al., 2001*). With regard to stress research, the nervous and endocrine systems regulating biological and behavioral responses to stress are highly conserved in zebrafish (*Stewart et al., 2012*). For example, the hypothalamic-pituitary-interrenal (HPI) axis of zebrafish is analogous to the HPA axis of mammals (*Nesan & Vijayan, 2013*; *Wendelaar Bonga, 1997*). When zebrafish encounter stress, similar to mammals, the hypothalamus of the animal releases corticotropin releasing hormone (CRH) to stimulate the release of adrenocorticotropic hormone (ACTH) from the anterior pituitary. ACTH stimulates the production and release of cortisol into the circulation from the interrenal cells (for a review of the zebrafish HPI axis, see *Alsop & Vijayan, 2009a*). Thus, the ability to measure basal and stress-induced cortisol neuroendocrine responses mirrors mammalian responses and is a major advantage of the zebrafish model.

In addition, zebrafish are not exempt from the physiological and behavioral adaptations in response to variable stressors, and different testing paradigms have been used to investigate behaviors characteristic of anxiety. Among these models, the novel tank test has been a powerful tool in analyzing habituation and motor activity (*Raymond et al., 2012*; *Wong et al., 2010*). In conditions where zebrafish have not been exposed to factors that may induce anxiety-like behavior, fish generally increase exploratory behavior and decrease freezing across the 6 min of the testing period (*Cachat et al., 2010*). Certain swimming patterns characteristic of anxiety, such as a decrease in time that the zebrafish spends in the top half of the tank, a decrease in the number of times the zebrafish enters

the top, and an increase in latency to enter the upper half have been observed as a result of anxiogenic factors (*Cachat et al., 2010*). However, a preference for the bottom can depend on the transparency of the tank (*Blaser & Rosemberg, 2012*). Anxiolytic agents, on the other hand, produce opposite effects in the novel tank test, such as a decreased latency to enter the top and increased entries to top (*Egan et al., 2009*).

Another complementary testing paradigm for anxiety-like behavior is the light/dark preference test (*Kysil et al., 2017*), in which the zebrafish is placed in a tank that typically consists of an equally divided light (or white) and dark side. The light/dark preference test is used to demonstrate scototaxis, or the preference of a dark compartment over a light compartment, as a model of anxiety (*Araújo et al., 2012*; *Champagne et al., 2010*). This test examines the instinctive motivation of the zebrafish to spend significantly more time in the dark side of the tank, as a means to protect itself from potential predators, over its innate behavior to explore a new environment (*Maximino et al., 2010*). The zebrafish is initially placed in the light compartment and allowed to freely explore the tank; during this time, the time spent in the light side, midline crossings, and distance traveled in the light side can be measured. Generally, zebrafish show an initial preference for the dark compartment when illumination above the tank is kept constant (*Facciol et al., 2019*; *Facciol, Tran & Gerlai, 2017*). Anxiolytic drugs tend to decrease the time that the zebrafish spend in the dark compartment and increase exploratory behavior, while anxiogenic drugs increase time spent in the dark compartment and decrease exploratory behavior (*Magno et al., 2015*; *Steenbergen, Richardson & Champagne, 2011*).

Characterization of biochemical markers associated with behavioral changes is essential for elucidating possible neuroadaptations elicited by stress exposure. Investigations in this area will allow for a more complete understanding of possible vulnerability factors for stress-related conditions or provide potential targets for pharmacological treatment options. A previous report detailed neuroendocrine and neurochemical levels in zebrafish following an acute (30-second) net handling stressor and reported a time-dependent increase in levels of whole body cortisol with no change in brain levels of serotonin or dopamine in adult zebrafish (*Tran, Chatterjee & Gerlai, 2014*). However, anxiety-like behavior was not measured in any behavioral test in response to the acute net stress in the aforementioned study (*Tran, Chatterjee & Gerlai, 2014*). Despite evidence that zebrafish demonstrate anxiety-like behaviors in response to physical stress and chemical anxiogenic agents, such as an acute net chase or exposure to conspecific alarm pheromone (*Mezzomo et al., 2019*; *Mocelin et al., 2015*), it is unclear whether an acute net stressor will elicit behavioral modification in zebrafish as measured by the novel tank test or the light/dark preference test.

The aim of the current study is to determine whether an acute net stressor is sufficient to increase anxiety-like behavior and whole-body cortisol in young adult zebrafish. Young adult zebrafish (90 days post-fertilization) were subjected to a brief net handling stressor and subsequently exposed to the novel test tank or the light/dark preference test. We expected an observable increase in anxiety-like behaviors in both of the behavioral tests immediately after the acute stressor exposure, as well as an increase in whole body-cortisol in the young adult zebrafish 15 min after the acute stressor.

## METHODS

### Animals

Wild-type zebrafish (*Danio rerio*) were bred from a stock population originally obtained from Carolina Biological Supply (Burlington, NC). After fertilization, embryos were washed with system water and kept at room temperature in 250 ml beakers in embryo medium (50 embryos/100 ml medium) until hatching (*Westerfield, 2000*). Larval zebrafish (from hatching until 14 days post-fertilization (dpf)) were maintained at a density of 50 larvae/200 ml stagnant water at room temperature. Larval fish were fed twice daily with dried, commercially-available larval fish food, and were subject to gentle water exchanges once daily. On 15 dpf, fish were gently moved to the system, a two-shelf, stand-alone housing rack (Aquaneering, San Diego, CA, USA) with a slow drip. The drip was slowly increased every few days to acclimate the fish to a steady stream of water by 30 dpf. The juvenile fish were maintained in 1.8L tanks at a density of approximately 5–6 fish/L until approximately 90 dpf (the day of testing). There were no visual barriers between home tanks. The system was maintained on a 14:10 h light/dark cycle, water temperature of 26 $\pm$ 2 °C, and pH 7.4 $\pm$ 0.2. After 30 dpf, fish were fed once daily with commercially-available flake food. On the day of the experiment, the individual tanks housing the mixed-sex young adult fish (approximately 90 dpf) were removed from the system and moved to the experimental room. The fish were allowed to acclimate to the experimental room for one hour before testing. All experimental procedures were conducted between 9:00 a.m. and 1:00 p.m. local time. As fish are considered exempt species according to the U.S. Animal Welfare Act, this work did not require oversight by the Institutional Animal Care and Use Committee (IACUC) of the institution. However, all procedures involving the care and use of the animals were conducted according to established recommendations (*Harper & Lawrence, 2011*; *National Research Council, 2011*; *Westerfield, 2000*).

### Acute net stressor

Randomly selected individual fish in the experimental condition ($n = 20$) were netted out of the home tank and suspended in the net above the water for 30 s (*Tran, Chatterjee & Gerlai, 2014*; *Tran & Gerlai, 2015*). A separate control group ($n = 20$) was not exposed to the acute net stressor. Half of each treatment group were subsequently exposed to either the novel tank test (Experiment 1) or the light/dark preference test (Experiment 2). Each sample was assigned a number upon selection from the home tank; the corresponding treatments for each sample were not revealed until after the automated behavioral analysis and cortisol assays were conducted.

### Experiment 1: Novel tank test

After the acute net stressor (for fish in the stressed condition, $n = 10$) or directly from home tank (for fish in the control condition, $n = 10$), fish were individually netted and placed into a trapezoidal novel tank the same size and dimensions as the home tank (approximately 7 cm $\times$ 33 cm $\times$ 15 cm, Aquaneering part number ZT180T) for six minutes. The behavior of the fish were recorded and subsequently analyzed with BehaviorCloud motion-tracking software (https://www.behaviorcloud.com/, San Diego, CA, USA). Number of entries to

the top of tank, time spent in top (sec), distance traveled in the top (cm), time spent in bottom (sec), distance traveled in the bottom (cm), and immobility duration (sec) were used as markers of anxiety behavior. A fish demonstrating anxiety-like behavior is less likely to explore the top, more likely to stay in the bottom zone of the novel tank, and will demonstrate more freezing behavior. Total distance traveled (cm) and mean speed (cm/s) were measured to ensure the acute stressor did not compromise activity levels (*Cachat et al., 2010*). One sample from each group was excluded from behavioral analyses due to incomplete video files.

## Experiment 2: Light/dark preference test

After the acute net stressor (for fish in the stressed condition, $n = 10$) or directly from home tank (for fish in the control condition, $n = 10$), fish were individually netted and placed into a rectangular tank (approximately 15 cm × 30 cm × 20 cm) with a water depth of 10 cm for fifteen minutes. The dark side of the tank (sides and bottom) was covered with black plastic aquarium background and the other side was left uncovered, as modified from previously published procedures (*Magno et al., 2015*; *Maximino et al., 2010*). The behavior of the fish were recorded and subsequently analyzed with BehaviorCloud motion-tracking software (https://www.behaviorcloud.com/, San Diego, CA, USA). Number of entries to the light zone, total time spent in light zone (min), and immobility duration (sec) were used as markers of anxiety behavior. Total distance traveled in the light side of the tank (cm) and velocity (cm/s) were measured to ensure the acute stressor did not compromise activity levels. One sample from the control group was excluded from behavioral analyses due to an incomplete video file.

## Euthanasia

In order to measure stress-induced cortisol responses at the peak of the response (*Ramsay et al., 2009*; *Tran, Chatterjee & Gerlai, 2014*), fifteen minutes after introduction to the behavioral test, each fish was placed individually in a 50 mL beaker containing 0.1% (100 mg/L) clove oil in system water. Death was determined upon visual examination for cessation of opercular (gill) movement and non-response to tactile stimulation (*Davis et al., 2015*). The whole-body samples were gently dried, then stored in individual microcentrifuge tubes at −20 °C.

## Determination of whole-body cortisol

Whole-body samples were used for assessing levels of cortisol (*Cachat et al., 2010*; *Canavello et al., 2011*). The samples from experiment 1 and experiment 2 were extracted and determined in independent procedures. Briefly, whole-body samples were thawed and weighed, then homogenized in one ml ice-cold 25 mM PBS buffer. Diethyl ether (five ml) was added to the homogenates to extract the cortisol. After centrifugation, the organic layer containing the cortisol was transferred to a new test tube. The ether/centrifugation step was repeated twice; all ether layers from each sample were collected in a single tube. The samples from Experiment 1 were allowed to dry at room temperature under a fume hood until the volatile compounds evaporated; samples from Experiment 2 were dried

with a light stream of air under the fume hood. In both procedures, samples were dried until a yellow oil containing cortisol remained.

After the evaporation, one ml 25 mM PBS was added to the lipid-containing extract in each tube. To determine cortisol levels, a cortisol enzyme-linked immunosorbent assay (ELISA) was used (Salimetrics, State College, PA) as per the manufacturer's instructions. Cortisol levels were normalized to whole-body weight and are expressed as ng cortisol/g whole-body weight. For Experiment 1, three samples (one from the control group and two from the stress group) were removed from the cortisol analysis due to issues with the extraction procedure. In Experiment 2, one sample from the control group was removed from the analysis due to an extraction error.

## Statistics
The data are presented as group means and the standard errors of the mean (SEM). Overall behavioral measures and cortisol levels were analyzed by $t$-tests for independent means. The behavioral data was also analyzed as a function of time (one and three minute bins for the novel tank test and the light/dark preference test, respectively) and analyzed with a repeated-measures ANOVA with Greenhouse-Geisser sphericity correction if Mauchly's test of sphericity indicated a violation of the sphericity assumption. JASP software (https://jasp-stats.org/, Amsterdam, The Netherlands) was used for statistical analyses. A significance value of $p < 0.05$ was used as the criterion for a result to reach statistical significance.

# RESULTS

## Experiment 1: Behavioral measures in the novel tank test and whole-body cortisol levels were not altered in response to acute net stress in young adult zebrafish

### Motor activity
Young adult zebrafish exposed to an acute net stressor did not show any differences in the total distance traveled (cm) or mean speed (cm/s) in the novel tank test compared to control fish (Table 1). A $t$-test for independent means indicated no significant effect of acute stress for either the total distance traveled ($t(16) = -0.169$, $p = 0.868$) or mean speed ($t(16) = 1.497$, $p = 0.154$). When the total distance data was broken down into six 60-s bins (Fig. 1A) and analyzed with a repeated-measures ANOVA, there was no effect of treatment ($F(1, 16) = 0.029$, $p = 0.868$), no effect of time ($F(2.284, 36.546) = 1.246$, $p = 0.303$), and no interaction between treatment and time ($F(2.284, 36.546) = 0.933$, $p = 0.413$). For mean speed over the first six minutes after being introduced into the novel tank (Fig. 1B), there was no effect of treatment ($F(1, 16) = 2.242$, $p = 0.154$), a significant effect of time ($F(3.327, 53.225) = 7.907$, $p < 0.001$), and no interaction between treatment and time ($F(3.327, 53.225) = 0.741$, $p = 0.545$). Generally, the mean speed of the zebrafish decreased across the task, but there was no effect of treatment.

### Immobility (freezing)
According to a $t$-test for independent means, young adult zebrafish exposed to an acute net stressor did not show any overall differences in immobility across the six minutes of

**Table 1  Behavioral measures of zebrafish in the novel tank test.** Exposure to an acute net stressor did not significantly alter overall behavioral measures in the novel tank test (6 min) in young adult zebrafish compared to unstressed (control) fish.

| Variable | Control | | Acute net stressor | | t | df | p |
|---|---|---|---|---|---|---|---|
| | M | SEM | M | SEM | | | |
| Total distance moved (cm) | 763.28 | 217.98 | 810.26 | 171.83 | −0.169 | 16 | 0.868 |
| Mean speed (cm/s) | 6.27 | 0.75 | 4.88 | 0.55 | 1.497 | 16 | 0.154 |
| Total time immobile (s) | 172.17 | 37.25 | 118.53 | 34.55 | 1.056 | 16 | 0.307 |
| Number of entries to top | 9.67 | 2.30 | 12.56 | 3.52 | −0.688 | 16 | 0.501 |
| Total time in top (s) | 61.11 | 16.19 | 65.78 | 24.79 | −0.158 | 16 | 0.877 |
| Distance in top (cm) | 119.33 | 50.02 | 157.54 | 75.96 | −0.420 | 16 | 0.680 |
| Total time in bottom (s) | 298.68 | 16.19 | 293.91 | 24.87 | 0.161 | 16 | 0.874 |
| Distance in bottom (cm) | 643.95 | 199.84 | 652.72 | 150.89 | −0.035 | 16 | 0.972 |

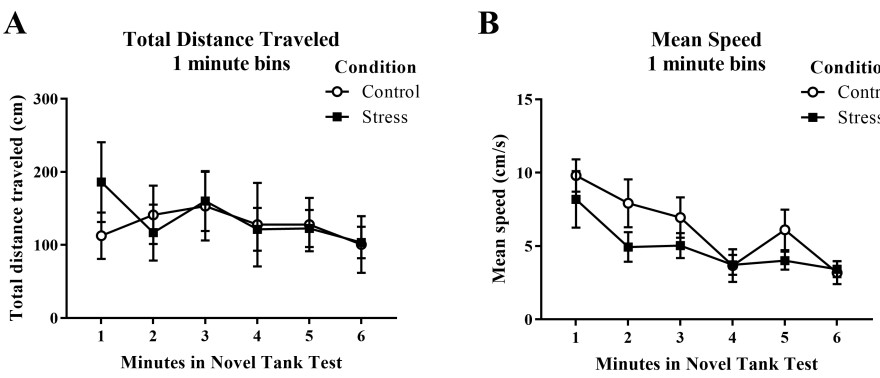

**Figure 1  Measures of zebrafish motor activity in the novel tank test over time.** Although the mean speed of young adult zebrafish generally decreased over time, acute net stress exposure did not alter (A) total distance traveled and (B) mean speed in the novel tank test. Bars indicate means of each group ± SEM.

the novel tank test (Table 1; $t(16) = 1.056$, $p = 0.307$). When immobility was analyzed by minute in the novel tank test (Fig. 2), there was no effect of treatment ($F(1, 16) = 1.115$, $p = 0.307$), a significant effect of time ($F(2.854, 45.667) = 4.998$, $p = 0.005$), and no interaction between treatment and time ($F(2.854, 45.667) = 0.637$, $p = 0.588$). The time that the zebrafish spent immobile decreased across the task, but there was no effect of treatment on this measure.

### Exploratory behavior

Young adult zebrafish exposed to an acute net stressor did not show any differences in time spent in the top zone (sec), the distance traveled in the top zone (cm), or the number of entries to the top in the novel tank test compared to control fish (Table 1). A $t$-test for independent means indicated no significant effect of condition for the time spent in the top zone ($t(16) = −0.158$, $p = 0.877$), the distance traveled in the top zone ($t(16) = −0.420$, $p = 0.680$), or number of entries to the top ($t(16) = −0.688$, $p = 0.501$). When the time

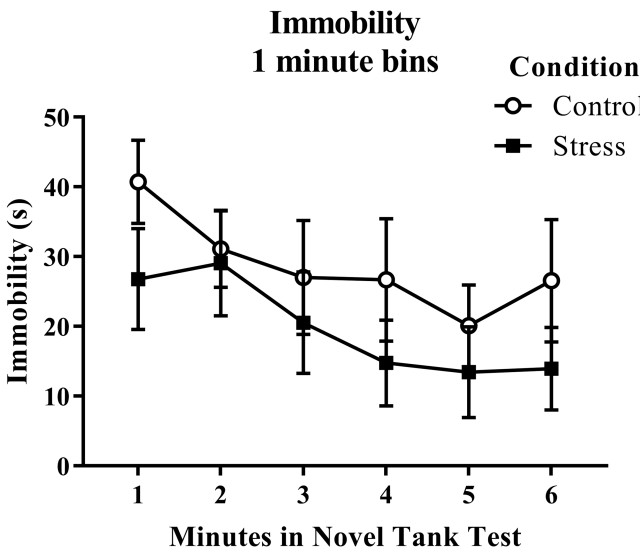

**Figure 2 Measure of zebrafish freezing behavior in the novel tank test over time.** Although the time spent immobile decreased over time, acute net stress exposure did not alter the immobility of young adult zebrafish in the novel tank test. Bars indicate means of each group ± SEM.

spent in the top zone was broken down into six 60-second bins (Fig. 3A) and analyzed with a repeated-measures ANOVA, there was no effect of treatment ($F(1, 16) = 0.025, p = 0.877$), no effect of time ($F(2.586, 41.382) = 0.677, p = 0.550$), and no interaction between treatment and time ($F(2.586, 41.382) = 1.871, p = 0.156$). For the distance traveled in the top zone (Fig. 3B), there was no effect of treatment ($F(1, 16) = 0.176, p = 0.680$), a significant effect of time ($F(2.522, 40.358) = 3.339, p = 0.035$), and no interaction between treatment and time ($F(2.522, 40.358) = 0.975, p = 0.402$). For the number of entries to the top zone (Fig. 3C), there was no effect of treatment ($F(1, 16) = 0.473, p = 0.501$), a significant effect of time ($F(5, 80) = 3.293, p = 0.009$), and no interaction between treatment and time ($F(5, 80) = 0.399, p = 0.848$). Generally, the fish explored the top zone of the novel tank less across the task, but there was no effect of treatment on upper zone exploration.

Young adult zebrafish exposed to an acute net stressor did not show any differences in time spent in the bottom zone (sec) and the distance traveled in the bottom zone (cm) (Table 1). A $t$-test for independent means indicated no significant effect of condition for the time spent in the bottom zone ($t(16) = 0.161, p = 0.874$) and the distance traveled in the bottom zone ($t(16) = -0.035, p = 0.972$). When the time spent in the bottom zone was broken down into six 60-second bins (Fig. 4A) and analyzed with a repeated-measures ANOVA, there was no effect of treatment ($F(1, 16) = 0.026, p = 0.874$), no effect of time ($F(2.589, 41.417) = 0.698, p = 0.538$), and no interaction between treatment and time ($F(2.589, 41.417) = 1.864, p = 0.158$). For the distance traveled in the bottom zone (Fig. 4B), there was no effect of treatment ($F(1, 16) = 0.001, p = 0.972$), no effect of time ($F(2.519, 40.310) = 1.584, p = 0.213$), and no interaction between treatment and time ($F(2.519, 40.310) = 1.919, p = 0.150$).

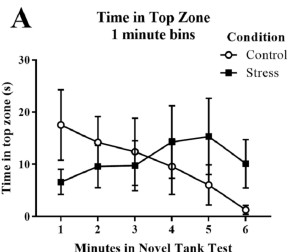
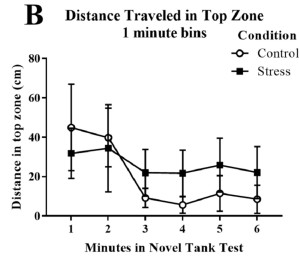
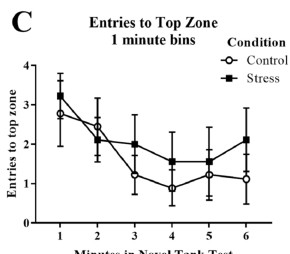

**Figure 3 Measures of zebrafish activity in the top zone of the novel tank test over time.** Although young adult zebrafish generally explored the top zone less over the time of the task, acute net stress exposure did not alter (A) the time spent in the top zone, (B) the distance traveled in the top zone, and (C) the number of entries to the top zone of the novel tank test. Bars indicate means of each group ± SEM.

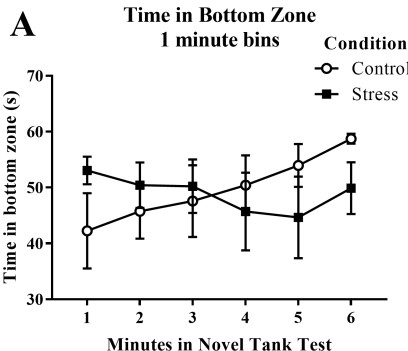
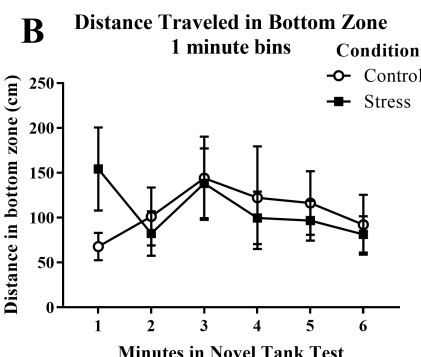

**Figure 4 Measures of zebrafish activity in the bottom zone of the novel tank test over time.** Acute net stress exposure of young adult zebrafish did not alter (A) the time spent in the bottom zone and (B) the distance traveled in the bottom zone of the novel tank test. Bars indicate means of each group ± SEM.

## *Whole-body cortisol*

Young adult zebrafish exposed to an acute net stressor and then subsequently placed in the novel tank test did not show any differences in whole-body cortisol levels compared to control fish (Fig. 5). A $t$-test for independent means indicated no significant effect of acute stress for cortisol levels ($t(15) = -0.079$, $p = 0.938$).

## Experiment 2: Behavioral measures in the light/dark preference test and whole-body cortisol levels were not altered by acute net stress in young adult zebrafish

### *Motor activity*

Young adult zebrafish exposed to an acute net stressor did not show any differences in the overall total distance traveled (cm) or mean speed (cm/s) in the light/dark preference test compared to control fish (Table 2). A $t$-test for independent means indicated no significant effect of acute stress for either the total distance traveled ($t(17) = -0.406$, $p = 0.689$) or mean speed ($t(17) = 0.094$, $p = 0.926$). When the total distance data was broken down into five 3-minute bins (Fig. 6A) and analyzed with a repeated-measures

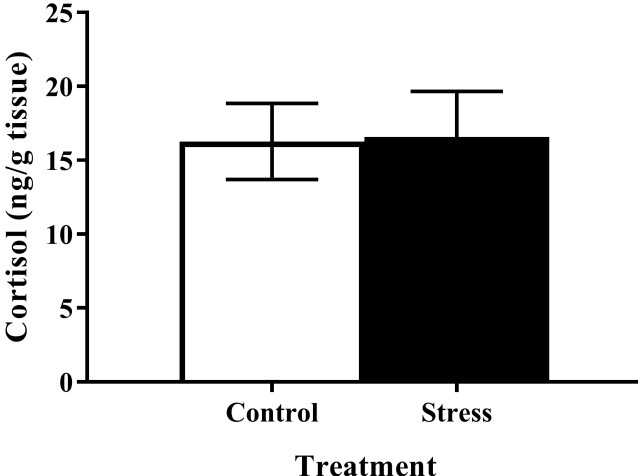

**Cortisol
(After Novel Tank Test)**

**Figure 5** **Measure of zebrafish neuroendocrine function after the novel tank test.** Acute net stress exposure did not alter whole-body cortisol levels of young adult zebrafish in Experiment 1 (fish were sacrificed after the novel tank test). Bars indicate means of each group ± SEM.

**Table 2** **Behavioral measures of zebrafish in the light/dark preference test.** Exposure to an acute net stressor did not significantly alter overall behavioral measures in the light/dark preference test (15 min) in young adult zebrafish compared to unstressed (control) fish.

| Variable | Control | | Acute net stressor | | t | df | p |
|---|---|---|---|---|---|---|---|
| | M | SEM | M | SEM | | | |
| Total distance moved (cm) | 2437.46 | 517.74 | 2131.06 | 542.21 | −0.406 | 17 | 0.689 |
| Mean speed (cm/s) | 8.66 | 0.97 | 8.80 | 1.13 | 0.094 | 17 | 0.926 |
| Total time immobile (s) | 48.06 | 24.29 | 42.69 | 13.09 | −0.200 | 17 | 0.844 |
| Number of entries to light zone | 123.00 | 18.67 | 102.90 | 14.82 | −0.852 | 17 | 0.406 |
| Total time in light zone (s) | 470.57 | 82.89 | 415.69 | 81.65 | −0.471 | 17 | 0.644 |

ANOVA, there was no effect of treatment ($F(1, 17) = 0.165$, $p = 0.689$), a significant effect of time ($F(2.614, 44.439)$, $p < 0.001$), and a significant interaction between treatment and time ($F(2.614, 44.439) = 2.943$, $p = 0.050$). For mean speed (Fig. 6B), there was no effect of treatment ($F(1, 17) = 0.007$, $p = 0.935$), a significant effect of time ($F(2.371, 40.307) = 6.747$, $p = 0.002$), and no interaction between treatment and time ($F(2.371, 40.307) = 2.039$, $p = 0.136$). Generally, the overall motor activity of the zebrafish changed across the 15-minute task, but there was no effect of treatment on the activity.

### Immobility (freezing)

Young adult zebrafish exposed to an acute net stressor did not show any differences in the time spent immobile in the light/dark preference test compared to control fish (Table 2). A $t$-test for independent means indicated no significant effect of acute stress on immobility

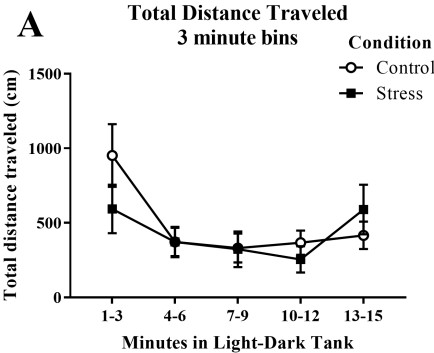

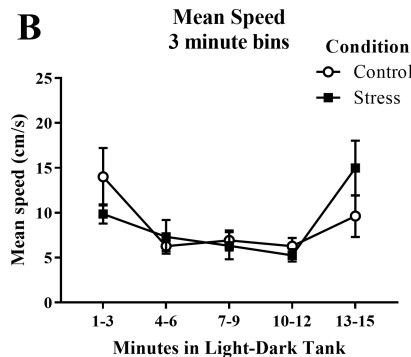

**Figure 6 Measures of zebrafish motor activity in the light/dark preference test over time.** Although the total distance traveled and mean speed of young adult zebrafish changed across the task, acute net stress exposure did not alter (A) total distance traveled and (B) mean speed in the light/dark preference test. Bars indicate means of each group ± SEM.

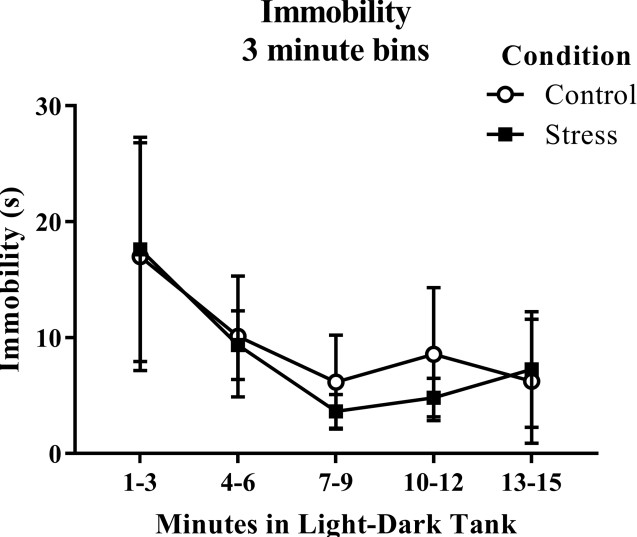

**Figure 7 Measure of zebrafish freezing behavior in the light/dark preference test over time.** Acute net stress exposure did not alter the immobility of young adult zebrafish in the light/dark preference test. Bars indicate means of each group ± SEM.

time ($t(17) = -0.200$, $p = 0.844$). When immobility was analyzed by 3-minute bins across the light/dark preference test (Fig. 7), there was no effect of treatment ($F(1, 17) = 0.040$, $p = 0.844$), no effect of time ($F(1.289, 21.906) = 2.097$, $p = 0.159$), and no interaction between treatment and time ($F(1.289, 21.906) = 0.090$, $p = 0.828$). The time that the zebrafish spent immobile decreased across the task, but there was no effect of treatment on this measure.
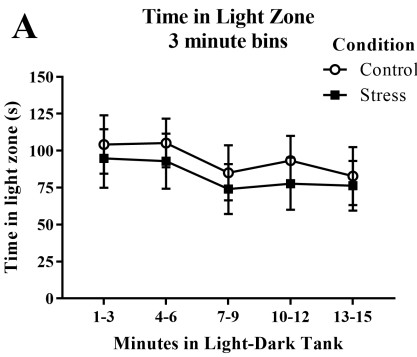
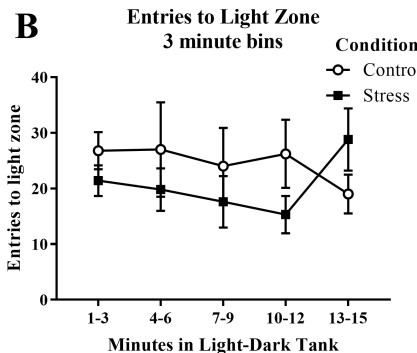

**Figure 8** **Measures of zebrafish activity in the light zone of the light/dark preference test over time.** Although the amount of time the young adult zebrafish spent in the light zone generally decreased over time, acute net stress exposure did not alter (A) the time spent in the light zone and (B) the distance traveled in the light zone of the light/dark preference test. Bars indicate means of each group ± SEM.

## *Exploratory behavior*

Young adult zebrafish exposed to an acute net stressor spent less time in the light zone in the light/dark preference test and entered the light zone fewer times compared to control fish (Table 2); however, these differences did not reach statistical significance ($t(17) = -0.471$, $p = 0.644$ and $t(17) = -0.852$, $p = 0.406$, respectively). When the time spent in the light zone was broken down into five 3-minute bins (Fig. 8A) and analyzed with a repeated-measures ANOVA, there was no effect of treatment ($F(1, 17) = 0.222$, $p = 0.644$), a significant effect of time ($F(4, 68) = 2.750$, $p = 0.035$), but no interaction between treatment and time ($F(4, 68) = 0.078$, $p = 0.989$). For the number of entries to the light zone (Fig. 8B), there was no effect of treatment ($F(1, 17) = 0.725$, $p = 0.406$), no effect of time ($F(2.224, 37.816) = 0.307$, $p = 0.760$), and no interaction between treatment and time ($F(2.224, 37.816) = 1.763$, $p = 0.182$). Generally, the fish explored the light zone of the light/dark tank less across the task, but there was no significant effect of treatment on light zone exploration.

## *Whole-body cortisol*

Young adult zebrafish exposed to an acute net stressor and then subsequently placed in the light/dark preference test did not show any differences in whole-body cortisol levels compared to control fish (Fig. 9). A $t$-test for independent means indicated no significant effect of acute stress for cortisol levels ($t(17) = 0.320$, $p = 0.753$).

## Conclusions

The goal of the current study was to demonstrate, in young adult zebrafish, that an acute net handling stressor would reliably increase whole-body cortisol levels, as was demonstrated in previous studies (*Tran, Chatterjee & Gerlai, 2014*; *Tran & Gerlai, 2015*). In addition, young adult zebrafish in the current study were immediately subjected to either the novel tank test or the light/dark preference test to assess the behavioral impact of the acute stressor. In contrast to previous research, the acute net stress was not sufficient to significantly alter whole-body cortisol levels 15 min after the acute stressor in young adult zebrafish. In

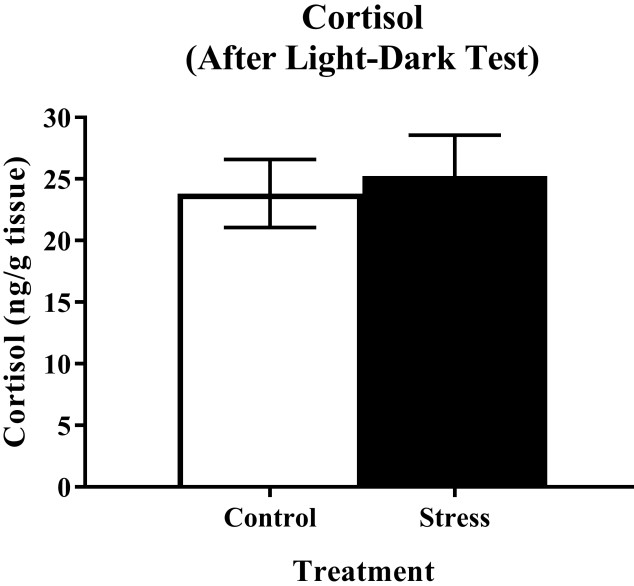

**Cortisol (After Light-Dark Test)**

**Figure 9** **Measure of zebrafish neuroendocrine function after the light/dark preference test.** Acute net stress exposure did not alter whole-body cortisol levels of young adult zebrafish in Experiment 2 (fish were sacrificed after the light/dark preference test). Bars indicate means of each group ± SEM.

support of this finding, there was no difference in anxiety-like behaviors as measured by the novel tank test and the light/dark preference test.

These findings may illustrate possible developmental differences in the stress response in zebrafish and may indicate that young adult zebrafish exhibit resistance to certain effects of mild stress. Studies in rodents demonstrate that the late adolescent period is marked by resiliency to certain behavioral effects of stress (*Jankord et al., 2011*). In humans, major changes in the neuroanatomy and sensitivity of certain brain structures, specifically in those structures that mediate responses to the external environment, occur during the adolescent period (*Andersen, 2003*). The lack of behavioral and neuroendocrine changes in response to stress in the current study potentially implies that, similar to other vertebrates, young zebrafish exhibit a degree of resiliency to certain stressors. Thus, a stronger stressor, such as confinement in a small tube (*Abreu et al., 2017*), may be required in order to elicit neuroendocrine and behavioral responses, at least in young zebrafish.

The lack of significant findings in the behavioral tests may be attributable to variations in experimental procedures and husbandry protocols rather than an absence of anxiety-like behavior elicited by acute stress in the young adult zebrafish. For example, in the initial published studies investigating the acute net stressor (*Tran, Chatterjee & Gerlai, 2014*; *Tran & Gerlai, 2015*), a week of isolation was conducted prior to the acute stressor. Because we were interested in investigating the practicality of eliciting stress responses of specifically this stressor in young adult fish, we did not replicate the isolation part of the methods. However, the results of the current study suggest the acute stressor was not sufficient to elicit stress responses when young adult zebrafish are housed in groups; thus, the effects of this specific acute stressor may only be observed in combination with previous social

isolation procedures. Other methodological differences may complicate the comparison of results between laboratories. For example, a recent study indicates that the frequency of feeding may impact the expression of anxiety-like behavior, with feeding once a day eliciting more anxiety-like behavior compared to feeding twice per day (*Dametto et al., 2018*). Thus, the results of the current study may have been affected by laboratory husbandry protocols. In addition, in the light/dark preference test, some labs use a central compartment in the testing tank and a 3-minute acclimation period (*Araújo et al., 2012*), whereas in the current study, a central compartment was not used and the zebrafish were initially netted into the light side. Another study used a water depth of 3 cm in the light/dark preference test (*Gebauer et al., 2011*), whereas the current experiment utilized a light/dark tank with 10 cm of water. Some studies leave the light compartment uncovered, while others cover it with an opaque or white material. Thus, these differences may mean that the light/dark test was not sensitive enough to capture anxiety-like behavior in the younger subjects.

Nevertheless, in the current study, the acute net stressor was tested in two separate experiments, using two behavioral tests, and neither the novel tank test nor the light/dark preference test produced observable behavioral modifications. In addition, the acute net stress did not elicit any observable changes in whole-body cortisol levels 15 min post-stressor in either of our experiments. It is entirely possible that young zebrafish have a different trajectory of whole-body cortisol release compared to adults (*Abreu De et al., 2014*; *Pavlidis, Theodoridi & Tsalafouta, 2015*; *Ramsay et al., 2009*; *Tran, Chatterjee & Gerlai, 2014*), with the peak occurring before or after 15 min post-stressor; however, the lack of effects observed in the behavioral tests strengthen the finding that the acute net stressor is not robust enough on its own to elicit stress responses in young adult zebrafish.

Overall, the current study provides additional information about the zebrafish as a model for studying physiological and behavioral effects of stress. Although many studies have investigated the effects brought upon by different stressors, literature in this field is typically limited to effects observed in either larval fish or during adulthood. Fewer studies have investigated stress regulation around the time of sexual maturation and in the young adult period (*Alsop & Vijayan, 2009b*; *Baiamonte et al., 2016*; *Dipp et al., 2018*; *Forsatkar et al., 2017*; *Petrunich-Rutherford, 2019*). Additional investigation of external factors and underlying mechanisms that mediate the effects of stress in the zebrafish can be used to further develop this animal as a practical model in neuroscience and further the current understandings of the plasticity and vulnerability of the stress response around the time of sexual maturation.

## CONCLUSIONS

This study was the first to determine that an acute net stressor is not sufficient to elicit anxiety-like behavioral responses in two different behavioral paradigms in young adult zebrafish. Contrary to expectations, the acute net stressor did not elicit increases in whole-body levels of cortisol in young adult zebrafish. Thus, a different stressor or a more intense or prolonged stressor may be necessary to elicit anxiety-like behaviors and neuroendocrine responses in this age group. Furthermore, additional investigations in the time course of

the stress-induced cortisol response of juvenile and young adult zebrafish are necessary to completely characterize stress responses in different developmental periods across the lifespan.

### Funding

This work was supported by the IU Northwest Faculty Grant-in-aid of Research and the Minority Opportunity for Research Experiences (MORE) program. The funders had no role in study design, data collection and analysis, decision to publish, or preparation of the manuscript.

### Grant Disclosures

The following grant information was disclosed by the authors:
IU Northwest Faculty Grant-in-aid of Research.
Minority Opportunity for Research Experiences (MORE) program.

### Competing Interests

The authors declare there are no competing interests.

### Author Contributions

- Amy Aponte conceived and designed the experiments, performed the experiments, analyzed the data, authored or reviewed drafts of the paper, approved the final draft.
- Maureen L. Petrunich-Rutherford conceived and designed the experiments, performed the experiments, analyzed the data, contributed reagents/materials/analysis tools, prepared figures and/or tables, authored or reviewed drafts of the paper, approved the final draft.

### Animal Ethics

The following information was supplied relating to ethical approvals (i.e., approving body and any reference numbers):

According to the Animal Welfare Act (AWA), this work did not require oversight by our Institutional Animal Care and Use Committee (IACUC) as it was not supported by Public Health Service (PHS) funding. In this case, fish are considered an exempt species (see Section 2132, part g of the AWA and letter from IUSM-IACUC, submitted as supplemental files). However, all procedures were still conducted according to ethical guidelines.

### Data Availability

The raw data are available in a Supplemental File. All behavioral measurements and whole-body cortisol levels for experiments 1 and 2 are provided.

### Supplemental Information

Supplemental information for this article can be found online at http://dx.doi.org/10.7717/peerj.7469#supplemental-information.

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
