# Peer review of "Acute net stress of young adult zebrafish (Danio rerio) is not sufficient to increase anxiety-like behavior and whole-body cortisol"

_PeerJ, doi:10.7717/peerj.7469_

## Round 0.1 · original submission · Major Revisions

The reviewers prepared several comments and queries for your attention, and I have attached a copy of this report below. I suggest to the authors read the PREPARE and/or ARRIVE guidelines to improve the description of the methodology used in the manuscript.

·

Basic reporting

The manuscript by Aponte and Petrunich-Rutherford investigates the effects of net acute stress on anxiety-like behavior and cortisol levels in young adult zebrafish. The data is clear, and the manuscript is nicely written. However, I have a few issues that need to be addressed and some concerns about the methods used.

Experimental design

I think that one of the major issues in the MS is the housing condition of the animals. In both works, the first one with adult animals (Tran and Gerlai, 2014; 2015) and this one with young adults, is that they do not discuss about the fact that zebrafish are isolated prior the stress and the behavioral task. Once that zebrafish is a social specie and that previously works showed that isolation changes zebrafish behavior (Parker et al., 2012), the authors should mention and discuss about this as an important variable and as a limitation of this work. In Tran works, they also moved the isolated the animals 7 days prior the test and, in this MS, animals were kept in this condition since 15 dpf. Further works aiming to analyze those differences should be very interesting, comparing about the differences in housing conditions vs stress reactivity, once that pair-house animals may have a different response compared to those that were kept isolated. Moreover, the authors should add more details about the housing condition in the Methods section (e.g. tank size, if animals had visual contact with the others).

(lines 134 – 136) Why the animals were fed once daily? I have some concerns that this could also affect zebrafish behavior, once that this is not part of standard zebrafish husbandry.

(lines 136 – 138) “On the day of the experiment, the individual tanks housing young adult fish (approximately 90 days post-fertilization) were removed from the system and moved to the experimental room”. The experimental room had the same temperature conditions from the fish room?

Although the authors mentioned that the light/dark task was used to assess zebrafish anxiety-like behavior, two recent papers have been showed that there are differences between the light/dark and white/dark test (Facciol et al., 2017; 2019). I think that is important the authors change the terminology used here and briefly comment about these differences in either Methods or Discussion section.

(lines 159 – 161) The authors said that one of the sides were covered with black and the other one was remained uncovered which is different from some protocols that cover the other side of the tank with white material to avoid contact with the external environment. This could also impact on zebrafish scototaxis responses.

(lines 189 – 192) “For the cortisol assays, a total of four samples (two samples from each treatment group) were removed from analyses due to issues with the extraction procedure. For the behavioral analyses, two samples from the non-stressed group were excluded from analyses due to incomplete video files”. However, in the excel files, the authors said that 3 animals were excluded for the cortisol analysis and 1 for the behavioral task, this only in the novel tank. For the light-dark more 1 animal was excluded from cortisol and other from behavioral analysis. Please, check the number of animals excluded and make the appropriate correction in the MS.

I noticed that the cortisol levels of the novel tank vs light-dark have a average difference of almost 10 ng/g. Where these animals from different batches? How many batches of fish the authors run?

Validity of the findings

The discussion section is well written and the conclusion well stated, however as previous mentioned, the authors should discuss about the limitations of their experiments, once that there are variations between this and previous experiments and is not only the age of the animals.

Additional comments

In the lines 67-78, should be interesting to mention about the cortisol production as one advantage of using zebrafish. Once that many others animal models do not produce this molecule.

(lines 82-85) The authors mentioned that in the novel tank animals usually spend half of the time in each zone, however previously works described the baseline behavior of zebrafish as initial increased preference for the bottom and then a gradually exploration for the top area (Cachat et al., 2010; Blaser and Rosemberg, 2012). Furthermore, in your data the animals showed in general a preference for the bottom of the tank, so I think that the authors could change this sentence and references to improve the rationale.

Please check the references, there are some places where the format is incorrect (e.g. Magno et al. (2015) - lines 102 and line 161).

·

Basic reporting

Graphs displaying some behaviours would helpful.
A schematic showing the behavioural testing designs and outlined behavioural zones (i.e. top) would be useful.

Experimental design

Is it possible that sample sizes may be too small? We normally consider n = 20 as appropriate for statistical power in behavioural zebrafish studies (this is specific to our lab but I could not find a simple article to support this claim). Could more samples be added?
Is euthanasia by clove oil too slow a method to accurately capture cortisol levels immediately after death? Why were fish not decapitated instead (as was the protocol in Tran et al. 2014)?
Could distance to top, instead of distance traveled in top zone, be a better measure? This may be a better behavioral measure of "bottom dwelling" (a well characterized anxiety-related behavior).

Validity of the findings

Why was an ANOVA not used to investigate a main effect of net stressor? It also would have been interesting to see the effect of the net stressor over time (repeated measures).

Additional comments

I do not think it is necessary to include how long the net stressor was every time it is mentioned. Simply explain in the methods that it was 30 seconds then just call it an acute net stressor. Continuously saying "acute (30 second) net stressor" makes the read a little hard at times, especially in the results.
Line 112 – “despite the evidence that zebrafish are demonstrate anxiety-like behaviors” please remove the “are”
Line 271 - “the lack of significant findings in the behavioural tests MAY be attributable...” please add "may" or similar word to clarify this sentence.

---

## Round 0.2 · accepted · Accept

Dear Dr. Petrunich-Rutherford
I am pleased to inform you that your manuscript referenced above has been accepted for publication in PeerJ.
Congratulations!
Kind regards,
Angelo

·

Basic reporting

English is very clear and the MS is well-written.

Experimental design

Research question is well defined, and the methods described had sufficient information to replicate.

Validity of the findings

The findings of this MS are very interesting and very important, especially from those scientists that works with zebrafish models.

Additional comments

The MS from Aponte and Petrunich-Rutherford was significantly improved, the new version has much more details about the methodology used and they have now addressed all the Reviewer concerns. The English is very clear, and the MS is well-written. Overall, I recommend this MS to be accepted in PeerJ.

·

Basic reporting

No comment

Experimental design

No comment

Validity of the findings

No comment

Additional comments

All previous comments have been addressed appropriately. I have reviewed the new manuscript and have no additional comments.